# Caffeine May Delay the Radiation-Induced Nucleoshuttling of the ATM Kinase and Reduce the Recognition of the DNA Double-Strand Breaks in Human Cells

**DOI:** 10.3390/biom16010041

**Published:** 2025-12-25

**Authors:** Léonie Moliard, Juliette Restier-Verlet, Joëlle Al-Choboq, Adeline Granzotto, Laurent Charlet, Jacques Balosso, Michel Bourguignon, Laurent Pujo-Menjouet, Nicolas Foray

**Affiliations:** 1Institut National de la Santé et de la Recherche Médicale (INSERM) U1296 Unit “Radiation: Defense, Health, Environment”, 28 Rue Laennec, 69008 Lyon, France; leonie.moliard@inserm.fr (L.M.); juliette.restier--verlet@inserm.fr (J.R.-V.); joelle.al-choboq@inserm.fr (J.A.-C.); adeline.granzotto@inserm.fr (A.G.); michel.bourguignon@inserm.fr (M.B.); 2Institute of Earth Science (ISTerre), Université Grenoble Alpes, Université Savoie Mont Blanc, Centre National de Recherche Scientifique (CNRS), Institut de Recherche pour le Développement (IRD), Université Gustave Eiffel, 38000 Grenoble, France; charlet38@gmail.com; 3Service de Radiothérapie, Centre Hospitalo-Universitaire (CHU) de Grenoble, 38000 La Tronche, France; jbalosso@wanadoo.fr; 4Université Paris Saclay, Université Versailles Saint-Quentin (UVSQ), 78180 Montigny le Bretonneux, France; 5Université Claude Bernard Lyon 1, Centre National de Recherche Scientifique (CNRS), Centrale Lyon, Institut National des Sciences Appliquées (INSA) Lyon, Université Jean Monnet, Institut Camille Jordan(ICJ) UMR5208, Institut National de Recherche en Sciences et Technologies du Numérique (INRIA), 69622 Villeurbanne, France; pujo@math.univ-lyon1.fr

**Keywords:** caffeine, radiation, ATM, DNA double-strand breaks, human fibroblast

## Abstract

Since 2014, a model of the individual response to ionizing radiation (IR), based on the radiation-induced nucleoshuttling of the ATM protein kinase (RIANS), has been developed by our lab: after irradiation, ATM dimers monomerize in cytoplasm and diffuse into the nucleus to trigger both recognition and repair of DNA double-strand breaks (DSB), the key-damage of IR response. Moderate radiosensitivity is generally caused by heterozygous mutations of ATM substrates (called X-proteins) that are over-expressed in cytoplasm and form complexes with ATM monomers, which reduces and/or delays the RIANS and DSB recognition. Here, we asked whether molecules, rather than X-proteins, can also influence RIANS. Caffeine was chosen as a potential “X-molecule” candidate. After incubation of cells with caffeine, cutaneous fibroblasts from an apparently healthy radioresistant donor, a patient suffering from Alzheimer’s disease (AD) and another suffering from neurofibromatosis type 1 (NF1) were exposed to X-rays. The functionality of ATM-dependent DSB repair and signaling was evaluated. We report here that caffeine molecule interaction with ATM leads to the inhibition of DSB recognition. This effect is significant in radioresistant cells. Conversely, in the AD and NF1 cells, the DSB recognition is already so low that caffeine does not provide any additional molecular effect.

## 1. Introduction

Since 2014, to better understand the individual molecular and cellular response to ionizing radiation (IR), our lab has proposed the radiation-induced ATM nucleoshuttling (RIANS) model [1]: shortly, IR cause radiolysis of water molecules, producing reactive oxygen species (ROS), and finally hydrogen peroxide. In the nucleus, this chemical process results in the formation of a specific number of base damage, DNA single- (SSB) and double-strand breaks (DSB), in proportion to the IR dose (about 10,000, 1000 and 40 per Gy per human cell, respectively) [2]. Concomitantly, such a chemical process triggers the dephosphorylation of the ATM protein kinase dimers, a major actor of the stress response [1]. The resulting ATM monomers (about 10^4^ to 10^6^ ATM monomers per Gy per cell [3]), present in cytoplasm, move towards the nucleus and diffuse across the nuclear membrane. Once in the nucleus, the ATM monomers phosphorylate the X variants of histone H2A (γH2AX), located at the DSB sites. This step is quantifiable by immunofluorescence through some bright spots (or foci) on the site of DSB [1]. The γH2AX foci reflect the radiation-induced DSB recognized by the ATM-dependent non-homologous end-joining (NHEJ) repair pathway, the most predominant DSB repair pathway in quiescent human cells [4]. As the DNA ends come together during the repair process, the ATM monomers phosphorylate and form ATM dimers. These ATM dimers are also quantifiable by immunofluorescence as foci, thanks to antibodies against the auto/trans-phosphorylated forms of ATM (pATM).

In the frame of the RIANS model, we have provided a relevant biological interpretation of the individual response to IR by defining 3 radiosensitivity groups (Figure 1) [1]. The group I is the group of radioresistant individuals whose cells recognize all the radiation-induced DSB in few minutes through a fast RIANS. They are all repaired by NHEJ (Figure 1A). The group II is the group of radiosensitive individuals at risk of cancer or else of degenerative disease [1] (Figure 1B). The group II individuals hold heterozygous mutations of genes that are generally phosphorylation substrates of ATM (6). These heterozygous mutations, instead of causing the disappearance of their gene products, are responsible for their overexpression and their abnormal localization in cytoplasm. These resulting proteins are called X-proteins. They form complexes with ATM monomers in the cytoplasm, which delays and/or reduces the flux of ATM monomers that diffuse into the nucleus. Consequently, not all the radiation-induced DSB are recognized by ATM via NHEJ and therefore not all the DSB are repaired (Figure 1). Lastly, the group III gathers the individuals associated with a gross impairment of the ATM-dependent NHEJ pathway associated with either the absence of DSB recognition or the absence of DSB repair. The group III is associated with hyper-radiosensitivity, very high risk of cancer or else of degenerative disease caused by homozygous mutations of genes [1,2]. While the hyper-radiosensitivity group III gathers the most severe phenotypes with the loss of gene function, the group II reflects moderate but significant radiosensitivity caused by a reduction and/or a delay of the RIANS via X-proteins (Figure 1B). More than 20 genetic diseases belonging to the radiosensitivity group II have been already characterized by the RIANS model and the corresponding X-proteins have been identified [1,5,6].

Interestingly, the RIANS model has been shown to be relevant for non-radiative environmental stress like pesticides and metals [7,8]. Hence, like X-proteins, one can hypothesize the existence of molecules that strongly interact with the ATM protein (in response to any given genotoxic stress) and reduce and/or delay the RIANS by interacting with the ATM monomers and preventing their diffusion in the nucleus (Figure 1C). Interestingly, the “X-molecules” hypothesis may rather reflect the impact of the environment on the IR response while the X-proteins likely depend on genetics: in order to investigate the relevance of the X-molecules hypothesis, we have deliberately chosen, as a first step, to use caffeine as an X-molecule candidate.

Coffee, and caffeine, its main active component, have been the subject of a plethora of molecular, cellular and clinical studies. The caffeine molecule [1,3,7-trimethylxanthine] belongs to the family of the alkaloid agents. The German Friedlieb Ferdinand Runge is considered as the first chemist to have extracted caffeine in 1819. While theine was isolated from tea in 1827 by the chemist Alfonse Oudry, Gerardus Mulder and Jobat demonstrated in 1838 that theine is actually the same molecule as caffeine [9,10]. In 1902, Hermann Emil Fischer elucidated the chemical structure of caffeine and achieved its first total synthesis [10]. Caffeine is a natural stimulant of the nervous system (psychotropic), and a secondary metabolite found in various products such as coffee, tea, chocolate, and some energy sodas. Due to its high daily consumption, it is the most frequently ingested psychoactive drug [9,10]. At the cellular scale, caffeine is known to deregulate the cell cycle, notably by inhibiting G2/M arrest, which favors cellular proliferation. Conversely, at the molecular scale, the specific properties of caffeine remain to be more documented, especially when combined with an exposure to IR [11,12].

After incubation of cells with caffeine, one group I and two group II human fibroblasts were exposed to X-rays: the functionality of ATM-dependent DSB repair and signaling was evaluated. We reported here, for the first time to our knowledge, that caffeine can be considered as an X-molecule in the frame of the RIANS model.

## 2. Materials and Methods

### 2.1. Cell Lines

All the experiments were performed with human untransformed fibroblasts that were routinely cultured at 37 °C in 5% CO_2_ humid conditions as monolayers with Dulbecco’s modified Eagle’s minimum medium (DMEM) (Gibco-Invitrogen-France, Cergy-Pontoise, France), supplemented with 20% fetal calf serum, penicillin, and streptomycin. All the experiments were performed with cells in the plateau phase of growth to avoid any cell cycle effect [6]. Three fibroblast cell lines were used in all the experiments: MRC5, providing from a 14 weeks old male foetus that generally served as a radioresistant control (group I) [4], and purchased from the European Collection of Authenticated Cell Culture (ECACC, Health Security Agency, Salisbury, UK); AG06840, providing from a patient suffering from Alzheimer’s disease (AD), who belongs to the group II of radiosensitivity [13], and purchased from the Coriell Institute (Camden, NJ, USA); RACKHAM37 (or R37), providing from a patient suffering from Neurofibromatosis type I (NF1), who belongs to the group II of radiosensitivity [6]. The NF1 cell line belongs to the abundantly documented COPERNIC collection, approved by a national ethical committee described elsewhere [4,14]. The resulting cells were declared under the numbers DC2008-585, DC2011-1437, and DC2021-3957 to the Ministry of Research. This collection obeys the French regulations about anonymous sampling and informed consent.

### 2.2. Treatment with Caffeine

Cells were pre-treated at concentrations of 0.1, 1, 3, 5, 10 mM caffeine (#C07550, Sigma-Aldrich, L’Isle-d’Abeau-Chesnes, France), by incubating caffeine directly to the culture medium for the indicated times. Such treatment was inspired by literature [15].

### 2.3. Irradiation

All the irradiations were performed on a 6 MeV X-ray clinical irradiator (SL 15 Philips) at the Anti-Cancer Centre Léon Bérard (Lyon, France) at a dose of 2 Gy with a dose rate of 6 Gy/min. Dosimetry was certified by the Radiophysics Department of Centre Léon Bérard.

### 2.4. Immunofluorescence

Immunofluorescence protocol and nuclear protein foci scoring were described elsewhere [16,17]. Anti-*γH2AX*^ser139^ antibody (clone JBW301; Merck, Millipore, Darmstadt, Germany) was applied at 1:800. Anti-*pATM*^ser1981^ (clone 10H11.E12; Millipore, Germany) were used at 1:100. The 4′,6-Diamidino-2-Phenylindole, Dihydrochloride (DAPI) counterstaining permitted to identify nuclei. Foci scoring procedure applied here has received the certification agreement of CE mark and ISO-13485 quality management system norms (www.iso.org) but developed some features protected in the frame of the Soleau Envelop and patents (FR3017625 A1, FR3045071 A1, EP3108252 A1). For all the experiments, a relative error of less than 7% for the larger number of foci (10 min and 1 h post-irradiation times) and less than 3% for the lower numbers of foci (4 and 24 h post-irradiation times) was considered. Such relative errors were found incompressible (a foci scoring based on 2, 5 or 10 times more nuclei did not decrease them. It is noteworthy that the size of nuclear γH2AX foci varies with post-irradiation time and their number per nucleus: the larger the number of foci, the smaller their size. Only the γH2AX foci with a size larger than 1 µm^2^ were considered [17].

### 2.5. In Situ Proximity Ligation Assay (PLA)

The proximity ligation assay (PLA) is a specific immunofluorescence technique that allows visualization of endogenous protein-protein interactions at the molecule scale, and the protocol has been described elsewhere [18]. Briefly, mixtures of two primary antibodies incubations were performed for 1 h at 37 °C, Anti-*ATM* (#OAEF00486, Aviva Systems Biology Corporation, San Diego, CA, USA) were used at 1:100 and anti-caffeine (#ARG42618, Arigo biolaboratories, Hsinchu County, Taiwan) were used at 1:200. The following antibodies were all diluted in the duolink antibody diluent 1X (#DUO82008, Sigma-Aldrich). PLA probes (Duolink PLA Probe anti-mouse MINUS #DUO82004-100RXN, Lot #SLCD468 and Duolink PLA Probe anti-rabbit PLUS #DUO82002-100RXN, Lot #SLLC564 from Sigma-Aldrich) were diluted using duolink antibody diluent at a ratio 1:5 and cells were incubated with the probes for 1 h at 37 °C in a humidified chamber.

### 2.6. Statistical Analysis

All the data were obtained with the indicated number of replicates, and the corresponding standard error of the mean was provided systematically. Two-way ANOVA to compare two numerical values and Spearman’s test to compare two kinetics were used and the corresponding *p* values were provided. It is noteworthy that the “null hypothesis” for these tests is that lower two-way ANOVA *p* values support significant differences between experimental data and higher Spearman’s test *p* values support the absence of significant differences between curves. Statistical analysis was performed by using Kaleidagraph v4 (Synergy Software, Reading, PA, USA).

## 3. Results

We first asked whether caffeine is a DSB inducer by treating cells with caffeine. Secondly, we examined the impact of caffeine on the recognition and repair of DSB induced by X-rays. Thirdly, we investigated the specificity of the interaction of caffeine to ATM. To these aims, the anti-*γH2AX* and anti-*pATM* immunofluorescence techniques were applied to a radioresistant (group I) fibroblast cell line derived from an apparently healthy individual and two radiosensitive (group II) fibroblast cell lines derived from patients suffering from Alzheimer’s disease (AD) (aging proneness) and neurofibromatosis type I (NF1) (cancer proneness). These last two cell lines have been deliberately chosen because (1) their major radiobiological features have been documented [6,13]; (2) the impact of caffeine in AD is of interest for neurologists [19], and (3) caffeine favours cellular proliferation and maybe the carcinogenesis process in cancer-prone diseases like NF1 [20].

### 3.1. Is Caffeine a DSB Inducer?

As a first step, anti-*γH2AX* immunofluorescence was applied to the three fibroblast cell lines maintained in quiescence (90–95% in G0/G1 phase) without any treatment, in order to assess their respective level of spontaneous *γH2AX* foci: an average of 0.3 ± 0.1 γH2AX foci per cell was found in the radioresistant control MRC5 cells, in agreement with our historical data concerning each of the three cell lines tested [6,13,21]. In the AD and NF1 fibroblasts, the number of spontaneous γH2AX foci was found to be 1.6 ± 0.5 and 0.8 ± 0.3 γH2AX foci per cell, respectively i.e., significantly higher than those of controls (*p* < 0.03). Again, these data are in agreement with published ones [6,13,21].

Cells were then treated to 0.1 to 10 mM caffeine for 15 min. The number of γH2AX foci increased with caffeine concentration for all the three cell lines tested but each, at a specific rate (Figure 2A,B). The radioresistant MRC5 controls showed 2–3 times less γH2AX foci than the two other cell lines; at 10 mM caffeine, the radioresistant MRC5 control showed 2.2 ± 0.1 γH2AX foci per cell while the two other cell lines showed 3 times more foci (*p* < 0.001). The slope of the curve made by data from the radioresistant control cell line was also found 3 times lower than those from the two other cell lines (*p* < 0.005). At 10 mM caffeine, the amounts of spontaneous γH2AX foci assessed in AD and NF1 cells were impressive: they were both higher than 6 γH2AX foci per cell on average. Let’s remind that, in our hands, an yield of more than 2 γH2AX foci per cell is deleterious for cells [21] (Figure 2A,B). Furthermore, it must be stressed that such values were clearly larger than spontaneous rates (i.e., without caffeine treatment) (*p* < 0.01) (Figure 2A,B).

The experimental protocol based on an incubation of cells with 0.1–10 mM caffeine for 15 min to cells in 90–95% quiescence cannot change significantly the cell cycle distribution. Hence, the DSB assessed after caffeine pre-treatment cannot be due to a significant population of cells entering in S-G2/M. In addition, very high concentrations of caffeine may inhibit some DSB repair pathways like NHEJ [22]. However, the number of spontaneous γH2AX foci was much lower than the number of γH2AX foci assessed at 10 mM caffeine, suggesting that the DSB assessed after caffeine pre-treatment cannot be the result of spontaneous DSB that would be become non-repairable and persistent in presence of caffeine. The γH2AX foci assessed after caffeine treatment strongly suggest the formation of additional DSB in presence of caffeine. Although impressive in AD and NF1 cells, such induction was found negligible in group I cells. Let’s remind that one coffee expresso represents about 11 mM caffeine (Figure 2B). However, in the case of an ingestion, the number of cells “physically” concerned by the contact with caffeine is likely to be very low, which suggests a limited biological and clinical impact (Figure 2B). It is noteworthy that the data obtained in the same condition with the anti-*pATM*-immunofluorescence reached the same conclusions (Figure 2C).

In order to compare the data obtained with caffeine treatment with some other stress, we plotted the data shown in Figure 2B with some historical data obtained in our lab with the same group I fibroblast MRC5 cell line incubated for 15 min with three metal species (Na_2_CrO_4_, AlCl_3_ and CuSO_4_) (Figure 2D) [7]. Interestingly, while the extreme toxicity and the property of chromium to induce DSB are well documented, the Figure 2D showed that the metal species tested induce also DSB but not in the same range of concentration (nM with chromium; μM with aluminium and copper; mM with caffeine) [7]. By expressing the number of γH2AX foci as Gy-equivalent (1 Gy corresponding to 40 γH2AX foci), it appeared that 10 mM caffeine applied to AD and NF1 cells induce the same amount of DSB as an exposure to 0.2 Gy X-rays. The dose-equivalent for group I cells was found to be about 0.05 Gy (Figure 2D). Again, the number of cells “physically” concerned by the contact with caffeine is likely to be very low, because of the successive dilutions in the body, which suggests a limited biological and clinical impact. Further experiments are needed to investigate the effects of chronic caffeine treatments.

### 3.2. Does Caffeine Impact on DSB Recognition and Repair?

To obtain DSB repair kinetics in caffeine-treated cells, the caffeine pretreatment was followed by an exposure to 2 Gy X-rays that induce a very documented and reproducible amount of DSB [21] (Figure 3). The γH2AX and the pATM foci were scored from 10 min to 24 h post-irradiation. Again, with this experimental protocol, quiescent untransformed fibroblasts incubated with caffeine for 15 min and exposed to a dose of 2 Gy X-rays cannot change their cell cycle distribution significantly.

The radioresistant group I fibroblasts, 10 min after 2 Gy X-rays and without the caffeine pre-treatment, generally showed 80 γH2AX foci at 10 min post-irradiation, which corresponds to the very documented DSB induction rate of 40 DSB per Gy per human cell [21]. It must be stressed here that the number of X-rays-induced DSB is much larger than that of spontaneous DSB (few DSB) and also than that of caffeine-induced DSB (up to 6 DSB). Hence, an exposure to Gy X-rays permits the evaluation of the effect due to caffeine to radiation-induced DSB (Figure 3A).

The number of radiation-induced γH2AX foci decreased continuously with repair time up to a zero value at 24 h post-irradiation. The DSB repair kinetics of the group I cells were in found in agreement with our very documented data [21]. After 1–5 mM caffeine pre-treatment, the number of initial γH2AX foci (10 min) was found significantly lower than non-irradiated controls (*p* < 0.001) and cannot be explained numerically by spontaneous data or caffeine-induced DSB (see below). At 24 h post-irradiation, the number of γH2AX foci were found higher than data without caffeine, only after a 5 mM caffeine pre-treatment. However, for all the caffeine concentrations tested, there was no statistical significance (*p* = 0.06) (Figure 3A).

Hence, 1–5 mM caffeine pretreatment may impair significantly the DSB recognition of the radioresistant control fibroblasts but without affecting significantly DSB repair. Since the number of caffeine-induced DSB is negligible by comparison to the number of radiation-induced DSB, our data suggest that caffeine affects significantly the recognition of the radiation-induced DSB by NHEJ. However, the ATM-recognized DSB did not show a slower DSB repair rate (Figure 3A).

When the same treatment was applied to the two group II cell lines characterized by a delayed RIANS, it appeared that the 1–5 mM caffeine pretreatment had no significant effect of the number of γH2AX foci assessed in GM06840 (*p* > 0.8) (Figure 3B) and in R37 fibroblasts (*p* > 0.7) (Figure 3C), suggesting that these two cell lines are resistant to caffeine after irradiation (Figure 3). Again, the same conclusions were reached with the pATM data (Figure 3D–F).

In order to analyse the kinetics data with another approach, the maximal numbers of γH2AX foci (10 min or 1 h) that reflect the maximal nuclear ATM kinase activity and therefore the largest DSB recognition power were plotted as a function of the caffeine concentration (Figure 4A). The group I cells data showed that the DSB recognition rate decreased significantly with caffeine pre-treatment until the minimal values reached with group II cells (*p* < 0.001). By contrast, the two group II cells elicited a maximal number of γH2AX foci ranging from 25 to 40 foci with non-significant change with caffeine concentration, suggesting a limited impact on DSB recognition (*p* > 0.5) (Figure 4A). With regard to the impact of DSB repair, the number of residual γH2AX foci reflecting DSB repair did not change significantly with caffeine concentration (*p* > 0.7) (Figure 4B). Again, the same conclusions were reached with the pATM data shown in Figure 4C,D.

### 3.3. Specificities of Cells Treated with Caffeine

During the experiments, we observed that a significant subset of cells pretreated with caffeine elicited abnormally high number of γH2AX foci: the highly damaged cells (HDC) are defined by more than 30 foci per cell. Interestingly, the number of HDC appeared to be a linear function of caffeine concentration (Figure 5A) but also of the corresponding γH2AX foci (Figure 5B) (at equal caffeine concentration).

Interestingly, unlike the intercept values, the slopes of the linear functions observed in Figure 5C were very similar for the three cell lines tested (*p* > 0.7): the percentage of HDC was found to be, on average, 10.9 times higher than the number of γH2AX foci assessed after caffeine pre-treatment (Figure 5C). This conclusion suggests that the formation of HDC is directly linked to the formation of DSB by caffeine.

### 3.4. Interaction Between ATM Protein and Caffeine Molecule

In order to ask whether caffeine and the ATM kinase interact, the proximity ligation assay (PLA) was applied to the same subset of cell lines with anti-*ATM* and anti-caffeine antibodies. Such assay consists in emitting light spot (dot) when ATM and caffeine are sufficiently close together to conclude to a physical interaction (Figure 6A). A negligible number of PLA dots was observed in cells without caffeine treatment. The number of PLA dots increased with caffeine concentration to reach an average value of 65 ± 3 PLA dots at 1 mM caffeine. This value was not found different from the three cell lines tested (*p* > 0.8). For higher caffeine concentrations, the number of PLA dots remained constant, increased or decreased in control, AD or NF1 cells, respectively (Figure 6B). The maximal values obtained for the three cell lines ranged from 60 to 80 dots per cell (Figure 6B).

In caffeine pre-treated cells that were irradiated at 2 Gy, 10 min post-irradiation, the number of PLA dots significantly decreased for all the cell lines tested (*p* < 0.01 at 5 mM caffeine). No significant difference was observed between the three curves (*p* > 0.8) (Figure 6C). These data suggest that a dose of 2 Gy X-rays may disturb the binding of the caffeine molecules to the ATM protein, while the caffeine concentration increases.

A PLA analysis was previously published for AD cells with the APOE protein as X-protein candidate and, similarly, for NF1 cells with the neurofibromin protein as X-protein candidate [6,13]. It must be stressed that 2 and 3 SQ/TQ domains specifically phosphorylated by ATM have been identified in APOE and neurofibromin proteins, respectively [6,13]. For the same cells tested, we have numerically compared the PLA values obtained with these X-proteins with those obtained with caffeine (Figure 6D). Interestingly, in control cells, the number of PLA dots indicating caffeine-ATM complexes was found systematically higher than those indicating the X-ATM complexes, suggesting that caffeine interacts spontaneously (without irradiation) more easily than the X-proteins tested, maybe due to favourable steric features and/or differences in the nature of the interaction. Furthermore, the number of PLA dots indicating caffeine-ATM complexes systematically decreased with irradiation while the number of PLA dots indicating X-protein-ATM complexes increased (Figure 6D). These findings may be the consequence of a radiation-induced over-expression of X-proteins generally observed in a number of genetic diseases tested with the RIANS model [6,13]. Conversely, the concentration of the X-molecules (here caffeine) does not increase with irradiation. In addition, irradiation may change the conditions of the interaction between ATM and X-molecules, likely with a change in the ATM protein conformation (monomerization, unwinding?) However, the diversity of interactions between X-molecules, X-proteins and ATM encourage us to investigate further the potential competition between X-molecules and X-proteins by new experiments.

## 4. Discussion

### 4.1. Caffeine, the Most Consumed Beverage in the World After Water

Among the numerous legends related to coffee origins, one of the most popular is that of the Ethiopian shepherd Kaldi, dating back to the 9th century after JC. This shepherd allegedly observed that his goats were livelier and more energetic after eating berries from a certain shrub. He then decided to taste it himself and experienced the same effects. Although this legend has not yet been historically proven, history shows that coffee may originate from Ethiopia, particularly from the town of Kaffa, where its trade developed throughout the Middle East until the 15th century. To date, coffee is the most consumed beverage in the world after water [9]. According to the US Food and Drug Administration (FDA), the daily caffeine consumption should not exceed 400 mg for most adults. Let us recall that a cup of coffee contains an average of 100 mg of this substance.

A famous example of an excessive consumption of coffee has been given by the French writer Honoré de Balzac. To stimulate his productivity and creativity, the writer consumed daily up to 50 cups of coffee to sustain the intense rhythm imposed by his work. It was said and believed that this abusive coffee consumption allegedly led to his premature death in 1850 at the age of 51 [9,23]. This story has highlighted the negative effects of unreasonable coffee consumption but also fed some beliefs about caffeine properties like severe dehydration, high addiction, heart pathologies, or even mental illnesses [24,25,26,27].

### 4.2. Caffeine, What We Know at the Cellular and Molecular Scales?

At the cellular scale, the effect of caffeine is dominated by its inhibition of cell cycle control. For radiobiologists, the G2/M checkpoint is particularly of interest since its dose-dependence is very documented [28,29,30,31,32]. This checkpoint is mediated by the cyclin B/CDK1 pair. IR induce the activation of the ATM kinase that phosphorylates different substrates involved in the following ordered and functional hierarchy: DNA damage recognition, DNA damage repair, cell cycle checkpoint and thereafter cell death pathways [33]. This is notably the case of Chk1 and Chk2 proteins, required for the G2/M and G1/S checkpoint, respectively [34]. The presence of caffeine prevents the interaction of ATM/ATR with Chk1/2, which is another argument that caffeine can bind to the ATM kinase. Indeed, our findings with PLA technique document a direct binding between caffeine and ATM may be applied to other kinases to support such specificity. By preventing the activation of checkpoint proteins, caffeine allows cells to continue dividing even in the presence of DNA damage, which could potentially lead to genomic instability and amplification of DNA damage like DSB [20].

At the molecular scale, caffeine (0.3–100 mM) applied directly to the culture medium, without irradiation, does not appear to induce a significant number of DSB to impact cellular lethality in radioresistant control fibroblast (Figure 2). Literature provides few evidence that caffeine may be a DNA strand break-inducer [12]. However, the mechanisms proposed to explain such capacity differ: notably, it was evoked that through the inhibition of cell cycle checkpoints by caffeine, some DNA single-strand breaks may be converted in DSB during the replication [12]. Furthermore, it has been hypothesized that high concentrations of caffeine inhibit the DSB repair pathways [22]. However, as specified in Results, the great majority of cells remained in G0/G1 and the additional DSB observed cannot be explained by any change in the cell cycle. In addition, the number of DSB reflected by the occurrence of γH2AX foci revealed a production of DSB proportional to the caffeine concentration. Lastly, the data related to HDC (Figure 5) strongly suggests that the occurrence of a phenomenon of amplification of DNA breaks, like hyper-recombination, as described elsewhere [35]. Hence, caffeine, at high concentration (> 1 mM) may be a DNA strand break-inducer but such level of DNA damage may not influence cellular lethality significantly in radioresistant control fibroblasts. Conversely, the case of AD and NF1 cell lines suggests that genomic instability (spontaneous DSB) increases the DSB potentially induced by caffeine molecules and favors the formation of HDC. Can a single model integrate the influence of caffeine on DNA breakage and how it inhibits cell cycle checkpoints?

### 4.3. The RIANS Model, One Single Model to Explain Two Properties of Caffeine

One of the most documented principles of radiobiology is, through the water radiolysis process, the dose-dependent production of oxygen peroxide (H_2_O_2_) molecules [36]. In the nucleus, H_2_O_2_ molecules are responsible for the DNA breakage through the DNA peroxidation. At high concentration, caffeine was shown to disturb the water radiolysis process, by inhibiting the expression of genes required for the anti-oxydation process like Nrf2 [37]. However, again, gene expression is a long process and may not be observed after 15 min caffeine pre-treatment. Our findings actually support a rapid chemical process. In fact, caffeine is also known to stimulate the Ca^2+^ ions release in response to a genotoxic stress [38]. Besides, such caffeine effect was found particularly efficient in bone cells, naturally rich in calcium [39]. The release of Ca^2+^ ions is one of the major biochemical events that follow an exposure to irradiation, known as the *bystander* effect and responsible for an excess of IR dose observed in targeted and non-targeted cells [40]. From all these assumptions, we can hypothesize that during the RIANS, the Ca^2+^-dependent release, stimulated by caffeine, may explain the induction of DSB and the presence of tiny γH2AX foci in cells [17,40]. In agreement with this hypothesis, the two group II cell lines tested (AD and NF1 fibroblasts), characterized by an intrinsic genomic instability and high yields of spontaneous DNA strand breaks showed higher number of caffeine-induced DSB (Figure 2). However, further experiments are needed to better document the bases of such assumptions and to investigate the role of Ca^2+^ ions in combination with caffein (Figure 7)

With regard to the DSB recognition, as evoked in Results, about 40 DSB per Gy per cell are generally observed in human fibroblast cells: 80 ɣH2AX foci were therefore expected after 2 Gy X-rays in control (group I cells) [1]: while the amount of spontaneous and caffeine-induced DSB was negligible by comparison with the amount of radiation-induced DSB, less than 40 ɣH2AX foci were found after a pre-treatment of 5 mM caffeine for 15 min followed by an exposure to 2 Gy X-rays. The PLA data suggested the existence of some caffeine-ATM complexes in cytoplasm that could explain a reduced and/or a delayed RIANS, supporting the lower amount of ɣH2AX foci: about 50% of radiation-induced DSB were not recognized by NHEJ.

With regard to DSB repair, we have already established correlations between the amounts of ɣH2AX and pATM foci, and cell survival at 2 Gy, quantifying cellular radiosensitivity. At 24 h post-irradiation, a non-negligible amount of unrepaired DSB was observed (about 4 ɣH2AX foci), suggesting a low toxicity due to 5 mM caffeine. However, this level was not found different from zero. The caffeine-induced DSB are not necessarily more severe than the radiation-induced ones. Again, further experiments are required to clarify the potential direct or indirect effect of caffeine on the DSB repair process of radiosensitive cells.

Lastly, the presence of X-proteins or caffeine on the ATM proteins may prevent its kinase activity, and notably the phosphorylation of CHK1 and CHK2 proteins that control G2/M and G1 arrests [41,42]. Further experiments are needed to document the effect of caffeine on DSB production DSB recognition and cell cycle checkpoints but a first mechanistic model can be proposed (Figure 7).

## 5. Conclusions

The biological action of caffeine is pleiotropic, strongly dependent on the concentration, and observed as various features at the molecular, cellular or clinical scale. Here, we gathered in the same mechanistic model the potential induction of DSB by caffeine, its influence of DSB recognition. These findings provide new variant of the RIANS model with X-molecules, instead of X-proteins, that may interact with ATM and prevent its nucleoshuttling. While further experiments are needed to better document this hypothesis and identify other X-molecules like caffeine, such a double model may permit to evaluate the contribution of genetics (X-proteins) and environment (X-molecules) in the individual post-stress response.

## Figures and Tables

**Figure 1 biomolecules-16-00041-f001:**
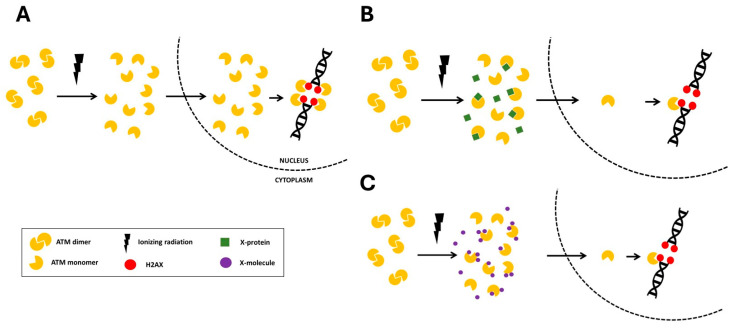
Schematic representation of the RIANS model. After irradiation, ATM dimers monomerize. The resulting ATM monomers diffuse into the nucleus. (**A**) In group I (radioresistant) cells, the RIANS is rapid and all the radiation-induced DSB are recognized by NHEJ. (**B**) In group II (radiosensitive) cells, the RIANS is reduced and slowed by the complexation of ATM with X-proteins. The DSB are not all recognized by NHEJ and therefore unrepaired. (**C**) In the frame of the “X-molecules” hypothesis, some drugs may directly interact with ATM protein and prevent the RIANS like with the X-proteins.

**Figure 2 biomolecules-16-00041-f002:**
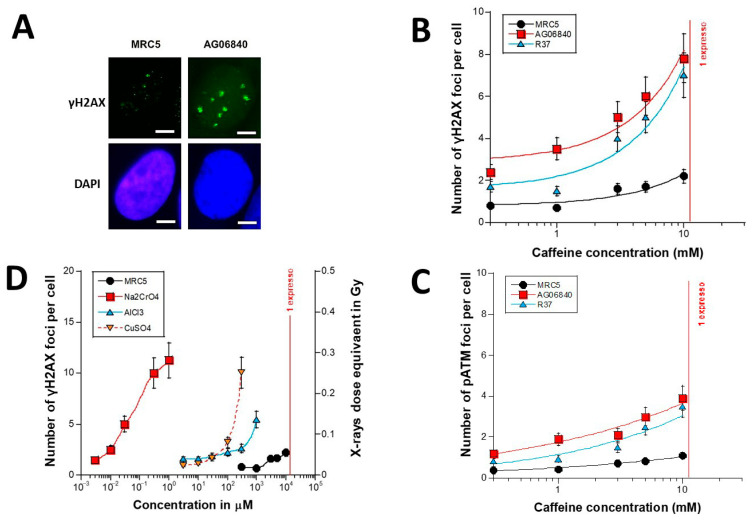
Number of γH2AX foci in cells treated to caffeine as a function of caffeine concentration. (**A**) Representative anti-γH2AX immunofluorescence images of fibroblasts treated to 10 mM caffeine for 15 min (green) and their corresponding DAPI counterstaining (blue). White bar represents 5 µm. (**B**) γH2AX data from the indicated cell lines treated to 0.1–10 mM caffeine for 15 min (the MRC5 cell line belongs to the group I of radioresistance); both AG06840 and R37 cell lines belong to the group II of radiosensitivity. The number of γH2AX foci shown is the mean ± standard error of the mean (SEM) of at least 2 independent replicates. (**C**) Number of pATM foci in cells treated to caffeine as a function of caffeine concentration. The pATM data from the indicated cell lines treated to 0.1–10 mM caffeine for 15 min. The number of pATM foci shown is the mean ± SEM of at least 2 independent replicates. (**D**) The data shown in Figure 2B were plotted together with some historical data obtained in our lab with the same group I fibroblast MRC5 cell line incubated for 15 min with three metal species (Na_2_CrO_4_, AlCl_3_ and CuSO_4_). The number of γH2AX foci was also expressed as X-rays dose equivalent based on 40 γH2AX foci per Gy. Red bar indicates the concentration of caffeine estimated in one expresso (about 11 mM caffeine).

**Figure 3 biomolecules-16-00041-f003:**
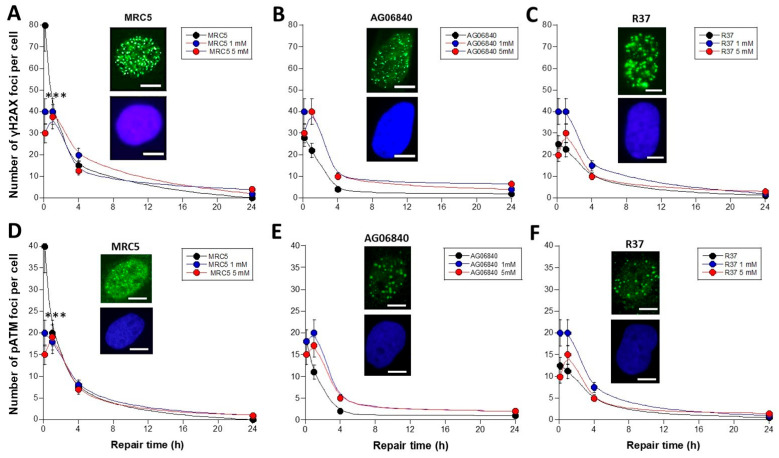
Number of γH2AX and pATM foci in cells treated to caffeine and X-rays as a function of time. The fibroblast cell lines ((**A**,**D**): MRC5; (**B**,**E**): AG06840; (**C**,**F**): R37) were treated to caffeine for 15 min at the indicated concentrations and exposed to 2 Gy X-rays. Anti-*γH2AX* or -*pATM* immunofluorescence was performed thereafter from 10 min to 24 h post-irradiation. The number of γH2AX or pATM foci remaining plotted as a function post-irradiation time is the mean ± standard error of the mean (SEM) of at least 2 independent replicates. Data were fitted to smooth function (see Materials and Methods). Inserts correspond to representative immunofluorescence images of cells at 10 min post-irradiation (green) with DAPI counterstaining (blue). White bar represents 5 µm. Three asterisks mean a significant difference with *p* < 0.001.

**Figure 4 biomolecules-16-00041-f004:**
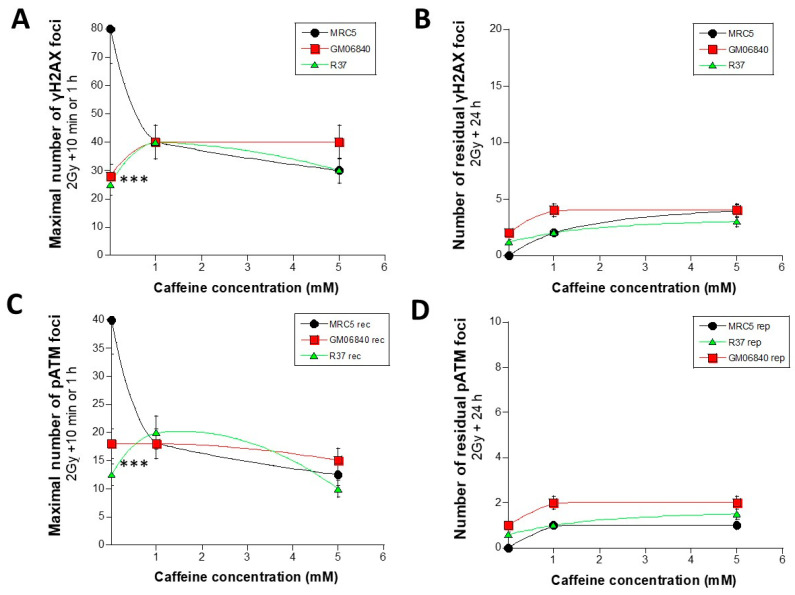
Impact of caffeine on the DSB recognition and repair rates. The maximal early number (10 min or 1 h post-irradiation) of γH2AX (**A**) and pATM (**C**) foci shown in Figure 3 was plotted against the caffeine concentration for the indicated cell lines. It reflects the DSB recognition rate. The number of γH2AX (**B**) and pATM (**D**) foci (shown in Figure 3) remaining at 24 h for repair was plotted against the caffeine concentration for the indicated cell lines. It reflects the DSB repair rate. For all the panels, the data plots correspond to the mean ± standard error of the mean (SEM) of at least 2 independent replicates. Data were fitted to a smooth fit. Three asterisks mean a significant difference with *p* < 0.001.

**Figure 5 biomolecules-16-00041-f005:**
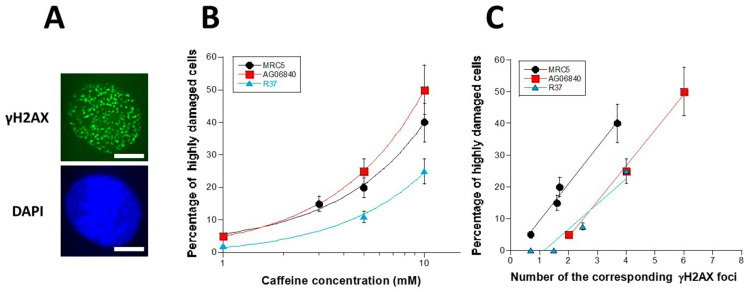
Percentage of HDC induced by caffeine. (**A**) Representative image of HDC (green) with DAPI counterstaining (blue). (**B**,**C**). The percentage of HDC cells was plotted against the caffeine concentration (**B**) and the number of corresponding γH2AX foci shown in Figure 2 at equal concentration (**C**). Each data plot represents the mean ± standard error of the mean (SEM) of at least 2 independent replicates. Data were fitted to a linear fit: y = −2.2 + 11.5x; y = −18.3 + 11.25x; y = −16.1 + 10.1x for the MRC5, AG08640 and R37 cell lines, respectively.

**Figure 6 biomolecules-16-00041-f006:**
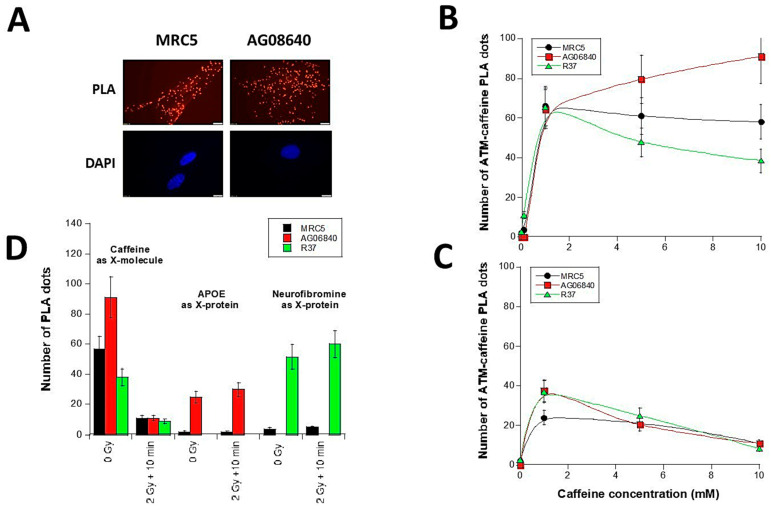
Interaction between caffeine and ATM. (**A**). Representative images of PLA dots obtained between the caffeine molecule and the ATM kinase protein (red) with DAPI counterstaining (blue). Data shown correspond to a treatment of 5 mM caffeine for the indicated cells. The number of the ATM-caffeine PLA dots has been plotted against the caffeine concentration for the indicated cell lines either spontaneously (**B**) or after 2 Gy X-rays followed by 10 min post-irradiation (**C**). Each data plot represents the mean ± standard error of the mean (SEM) of at least 2 independent replicates. Data were fitted to a curvilinear fit. (**D**). The number of the ATM-caffeine (from Figure 6B,C), ATM-APOE (from [13]) and ATM-neurofibromine (from [6]) PLA dots has been plotted together. Only data obtained from the spontaneous state (no radiation, 0 Gy) or after 2 Gy followed by 10 min were shown. Each data plot represents the mean ± standard error of the mean (SEM) of at least 2 independent replicates.

**Figure 7 biomolecules-16-00041-f007:**
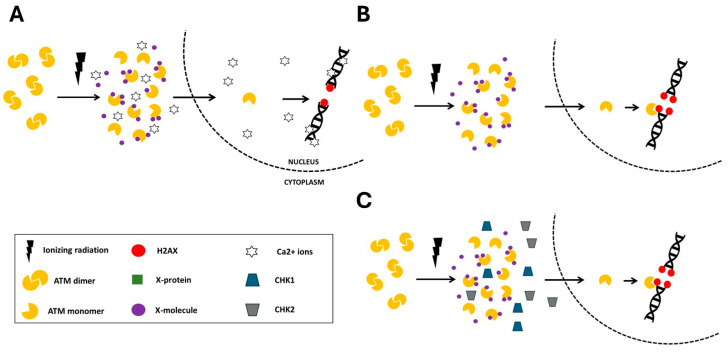
DSB induction, DSB recognition and inhibition of the cell cycle checkpoints in presence of caffeine in the frame of the same RIANS model. After irradiation, ATM dimers monomerize. (**A**) Irradiation and presence of caffeine stimulate the Ca^2+^ ions release that produces additional DSB and HDC. (**B**) Caffeine molecules interact with ATM monomers, which reduces and/or delays the RIANS (**C**) Caffeine molecules interact with ATM monomers, which reduces the probability of interaction between ATM and CHK1/2 proteins. Since these proteins are not phosphorylated by ATM, cell cycle checkpoints are not ensured.

## Data Availability

The original contributions presented in this study are included in the article. Further inquiries can be directed to the corresponding author.

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
