# Peer review of "Caffeine May Delay the Radiation-Induced Nucleoshuttling of the ATM Kinase and Reduce the Recognition of the DNA Double-Strand Breaks in Human Cells"

_biomolecules, 2025, doi:10.3390/biom16010041_

Round 1
Reviewer 1 Report
Comments and Suggestions for Authors
In this manuscript, the authors investigate the effect of caffeine treatment on the ATM-mediated cellular response to ionizing radiation. Using three fibroblast lines—one radioresistant control and two patient-derived—they monitor the number of γ-H2AX foci per cell as a readout of ATM activation in DSB signalling after X-ray exposure, with or without caffeine pretreatment. While the topic is potentially interesting and caffeine is indeed known to interfere with ATM signaling at multiple levels, the current version of the manuscript does not clearly address the mechanistic questions that are suggested in the abstract and introduction. As a result, the novelty and impact of the work remain uncertain.
The experiments presented focus mainly on γ-H2AX foci quantification and PLA assays to detect a possible interaction between caffeine and ATM. However, several central claims (the effects on DSB recognition and repair, ATM activity, and the mechanistic steps at which caffeine acts) are insufficiently supported by the data shown. Furthermore, certain conclusions appear inconsistent or overinterpreted.
Overall, the manuscript requires substantial revision, additional mechanistic experiments, and clarification of several points before it can be considered for publication.
Major comments:
- The main scientific question is not fully addressed. The abstract states that the authors aimed to determine “whether and how caffeine influences the different steps” of ATM activation, including dimer/monomer transition, nuclear translocation, and early signaling events. These aspects were not directly examined. The work is instead limited mainly to γ-H2AX foci quantification and PLA. To support mechanistic claims, some additional experiments are needed, such as the assessment of ATM protein levels and/or kinase activity, ATM subcellular localization, ATM dimer-to-monomer transition, evaluation of downstream targets beyond γ-H2AX,...
- The manuscript contains conflicting statements regarding DSB recognition and repair: at lines 30-31, the authors state that caffeine “leads to the inhibition of DSB recognition and repair.” while at lines 272-273, they conclude that caffeine “may impair significantly the DSB recognition … but without affecting significantly DSB repair.” Additional analyses of DSB repair could help to clarify this point (e.g., focus resolution kinetics, neutral comet assay, reporter assays, 53BP1 foci dynamics, ...).
- Figure 2 suggests that caffeine treatment induces DSB accumulation in patient-derived fibroblasts but not in healthy controls. However, the manuscript does not distinguish whether these foci arise from: direct DNA damage caused by caffeine or failure to repair endogenous DSBs. This point should be clarified or discussed.
- Figures 3 and 4. The data show reduced γ-H2AX formation immediately after irradiation in caffeine-treated healthy cells, but normal resolution thereafter. This suggests that caffeine impairs DSB detection rather than repair, contradicting the statement that DSB repair is inhibited (see point 2). Moreover, the biological significance of altered DSB detection is not addressed. The authors should examine whether caffeine affects cell viability, proliferation, clonogenic survival or checkpoint activation after irradiation to understand the impact of caffeine on DSB signaling.
- Lines 336–338 and Figure 6. The term “spontaneously” is ambiguous. Does it means “in the absence of irradiation” or “immediately after treatment,”?Additionally, the authors claim that similar PLA values were obtained using X-proteins with SQ/TQ domains phosphorylated by ATM (lines 347–348). These data should either be shown or referenced, as they are important for interpreting whether the PLA signal reflects ATM–caffeine interaction or non-specific binding to phosphorylated substrates.
- The discussion is excessively long and reads more like a general review of ATM and caffeine biology rather than an interpretation of the authors' own data. It should be revised.
Author Response
Reply to reviewer 1 :
We thank the reviewer 1 for his/her comments. The manuscript has been very deeply modified and new data and figures have been added (text highlight in yellow) to reach the requirements of the reviewer.
In this manuscript, the authors investigate the effect of caffeine treatment on the ATM-mediated cellular response to ionizing radiation. Using three fibroblast lines—one radioresistant control and two patient-derived—they monitor the number of γ-H2AX foci per cell as a readout of ATM activation in DSB signalling after X-ray exposure, with or without caffeine pretreatment. While the topic is potentially interesting and caffeine is indeed known to interfere with ATM signaling at multiple levels, the current version of the manuscript does not clearly address the mechanistic questions that are suggested in the abstract and introduction. As a result, the novelty and impact of the work remain uncertain.
The experiments presented focus mainly on γ-H2AX foci quantification and PLA assays to detect a possible interaction between caffeine and ATM. However, several central claims (the effects on DSB recognition and repair, ATM activity, and the mechanistic steps at which caffeine acts) are insufficiently supported by the data shown. Furthermore, certain conclusions appear inconsistent or overinterpreted.
Overall, the manuscript requires substantial revision, additional mechanistic experiments, and clarification of several points before it can be considered for publication.
Major comments:
- The main scientific question is not fully addressed. The abstract states that the authors aimed to determine “whether and how caffeine influences the different steps” of ATM activation, including dimer/monomer transition, nuclear translocation, and early signaling events. These aspects were not directly examined. The work is instead limited mainly to γ-H2AX foci quantification and PLA. To support mechanistic claims, some additional experiments are needed, such as the assessment of ATM protein levels and/or kinase activity, ATM subcellular localization, ATM dimer-to-monomer transition, evaluation of downstream targets beyond γ-H2AX.
- We agree. We have added pATM immunofluorescence data in supplementary data and their interpretation completes and consolidates the gH2AX ones. This was an enormous effort with regard to the time allowed. These pATM data are related to ATM activation and it is noteworthy that ATM expression is not necessarily correlated to ATM activity. Let’s recall that pATM antibodies detect only dimers while ATM antibodies detect both ATM dimers and monomers.
The reviewer should consider the twenty published articles or so that consolidate the model. Here, the major question raised in this paper : can the presence of caffeine modify the kinetics of gH2AX and pATM foci in human cells. The radiobiological features of cells used here have been also documented with regard to their molecular and celluar responses to radiation (see refs). All our data are in agreement with the very documented ATM nucleoshuttling model strongly supported by a mathematical frame in which constraints are important. Finally, the cellular events (notably cell cycle and cell death) are not in the scope of the manuscript : therefore, we did not investigate events downstream pATM and gH2AX foci formation. See new abstract, additional figures in Suppl data, modified figure in the manuscript and modified text highlighted in yellow.
- The manuscript contains conflicting statements regarding DSB recognition and repair: at lines 30-31, the authors state that caffeine “leads to the inhibition of DSB recognition and repair.” while at lines 272-273, they conclude that caffeine “may impair significantly the DSB recognition … but without affecting significantly DSB repair.” Additional analyses of DSB repair could help to clarify this point (e.g., focus resolution kinetics, neutral comet assay, reporter assays, 53BP1 foci dynamics, ...)You are right. We agree. We have modified the abstrat and correct the text in question: the influence on repair was not found significant.
It must be stressed here that the techniques proposed would add confusion in the data interpretation since we have documented their specific technical artefacts :
- Focus resolution : we have recently published in this journal a large paper about the artifacts related to the H2AX foci resolution (Granzotto et al., Biomolecules, 2024): this is not the scope of the paper but the materials and methods mentioned it.
- Neutral comet assay : it would mix (by the tail moment) the effect on chromatin (DNA breakage on the head of the comet) and the effect on repair (migration of DNA breaks in the tail). Such mixture would not add clearity in our data. Lastly, it is impossible to set up comet assay in 10 days allowed
- Reporter assays : it means that we need to transfect in normal fibroblasts some plasmids : considering the poor ratio of transfection obtained in primary fibroblasts, we should need transformed fibroblasts which would add biases in the data
- 53BP1 foci : we have fully documented that the kinetics fof 53BP1 foci is not the same in radiosensitive (-group II cells) as those with gH2AX foci, (Joubert et al. IJRB, 2008) suggesting that there is not overlapof both 53BP1 and gH2AX in the DSB sites : co-immunofluorescence would add confusion and biaises in the study (Renier et al., IJRB2007)
- See modified absract and additional (pATM) data
Figure 2 suggests that caffeine treatment induces DSB accumulation in patient-derived fibroblasts but not in healthy controls. However, the manuscript does not distinguish whether these foci arise from: direct DNA damage caused by caffeine or failure to repair endogenous DSBs. This point should be clarified or discussed.You are right. we agree : we have added spontaneous data before caffeine treatment to better distinguish between caffeine-induced DSB and endogenous. See modified text paragraph 3.1 line 205-223
Figures 3 and 4. The data show reduced γ-H2AX formation immediately after irradiation in caffeine-treated healthy cells, but normal resolution thereafter. This suggests that caffeine impairs DSB detection rather than repair, contradicting the statement that DSB repair is inhibited (see point 2). Moreover, the biological significance of altered DSB detection is not addressed. The authors should examine whether caffeine affects cell viability, proliferation, clonogenic survival or checkpoint activation after irradiation to understand the impact of caffeine on DSB signaling We have reinforced our data interpretation with regard DSB recognition and repair and endogenous DSB in the Results chapter. Again, it is said line 286 that : « Since the number of caffeine-induced DSB is negligible by comparison to the number of radiation-induced DSB, our data suggest that caffeine affects significantly the recognition of the radiation-induced DSB.”
However, it must be stressed again that the scope of our paper is the molecular events occurring before any influence of checkpoint or on cell survival. By the way, a rigorous series of cell survival experiments would require a number of preliminary experiments with regard the design of the pre-treatment (concentration, multiple addings of caffeine during the colony formation, etc). See modified text in the first two paragrpahs of Results.
Lines 336–338 and Figure 6. The term “spontaneously” is ambiguous. Does it means “in the absence of irradiation” or “immediately after treatment,”? Additionally, the authors claim that similar PLA values were obtained using X-proteins with SQ/TQ domains phosphorylated by ATM (lines 347–348). These data should either be shown or referenced, as they are important for interpreting whether the PLA signal reflects ATM–caffeine interaction or non-specific binding to phosphorylated substrates.Yes You are right : we have modified in all the manucript the text containing the « spontaneous » term. Like in any of our studies and other literature data, spontaneous means without treatment (=endogenous).
Yes. You are right. On the same cell lines (AD) and NF1) we have demonstrated in our previous reports with PLA assays that AD and NF1 cells are characterized by at least one specific X-protein (APOE and neurofibromin, respectively) in dedicated papers. See modified text and new figure 6D
The discussion is excessively long and reads more like a general review of ATM and caffeine biology rather than an interpretation of the authors' own data. It should be revised.OK see modified text. We have shorten the original text but , in addition to our explanations in results, we have provided a new reading of our model. See modified text in discussion and new figure 7.
Reviewer 2 Report
Comments and Suggestions for Authors
The article entitled “Caffeine delays the radiation-induced nucleoshuttling of the ATM kinase and reduces the recognition of the DNA double-strand breaks” is a research work specifically focused on the radiation-induced nucleoshuttling of the ATM kinase. Authors are encouraged to consider the following comments and suggestions for further refinement of their work
- The highest caffeine doses used seem much higher than what is found in the human body; the authors should better explain their choice.
- The claim that caffeine causes DNA breaks is based on one method; using another technique would make this finding stronger.
- The figures need clearer labels to show which results are statistically significant.
- The authors state the effect is likely small, but their data shows a real impact; this contradiction should be clarified.
- The experiment showing caffeine sticks to the ATM protein relies heavily on one specific antibody, which needs more validation.
- The explanation for how caffeine might cause DNA breaks is mostly a guess and should be stated more cautiously.
- The abstract's summary of the results in the Alzheimer's and NF1 cells is not fully supported by the data presented in the figures
Author Response
Reply to reviewer 2 :
We thank the reviewer 2 for his/her comments. The manuscript has been very deeply modified and new figures and new data have been added (text highlight in yellow) to reach the requirements of the reviewer.
The article entitled “Caffeine delays the radiation-induced nucleoshuttling of the ATM kinase and reduces the recognition of the DNA double-strand breaks” is a research work specifically focused on the radiation-induced nucleoshuttling of the ATM kinase. Authors are encouraged to consider the following comments and suggestions for further refinement of their work
- The highest caffeine doses used seem much higher than what is found in the human body; the authors should better explain their choice.
OK see modified text in Materials and methods and in the first paragraph of results
- The claim that caffeine causes DNA breaks is based on one method; using another technique would make this finding strongerTo reach the requirements of reviewer 1 .We have added pATM immunofluorescence datt in supplementary data and their interpretation completes and consolidates the gH2AX ones. This was an enormous effort with regard to the time allowed. These pATM data are related to ATM activation and it is noteworthy that ATM expression is not necessarily correlated to ATM activity. Let’s recall that pATM antibodies detect only dimers while ATM antibodies detect both ATM dimers and monomers.
The reviewer should consider the twenty published articles or so that consolidate the model. Here, the major question raised in this paper : can the presence of caffeine modify the kinetics of gH2AX and pATM foci in human cells. The radiobiological features of cells used here have been also documented with regard to their molecular and celluar responses to radiation (see refs). All our data are in agreement with the very documented ATM nucleoshuttling model strongly supported by a mathematical frame in which constraints are important. Finally, the cellular events (notably cell cycle and cell death) are not in the scope of the manuscript : therefore, we did not investigate events downstream pATM and gH2AX foci formation. See new abstract, additional figures in Suppl data, modified figure in the manuscript and modified text highlighted in yellow.
The figures need clearer labels to show which results are statistically significant.
OK see modified figures
The authors state the effect is likely small, but their data shows a real impact; this contradiction should be clarified.
Yes there was a confusion of the caffeine effect on DSB recognition and repair. We have reinforced our data interpretation with regard DSB recognition and how clearly that the effect of repair is not significant See modified text in abstract and in Results.
The experiment showing caffeine sticks to the ATM protein relies heavily on one specific antibody, which needs more validation.
The antibody against caffein used here was available and functional. We have add data related X-proteins which gives consistence to the interpretation of our data. See modified text in results and new figure Fig. 6D
The explanation for how caffeine might cause DNA breaks is mostly a guess and should be stated more cautiously.
OK . We have deeply modified the paragraphs of results and discussion related to this item See modified Results section 3.1. See also new figure 7.
The abstract's summary of the results in the Alzheimer's and NF1 cells is not fully supported by the data presented in the figures
OK see modified abstract and above comments
Round 2
Reviewer 1 Report
Comments and Suggestions for Authors
In this revised version of the manuscript the authors add some experiments and modify the text accordingly with my suggestion.
In particular, I appreciate that in this revised version the abstract has been effectively aligned with the actual content of the paper.
The manuscript has undergone substantial revisions, resulting in a much clearer and more focused presentation. Overall, the updated version represents a significant improvement in clarity, coherence, and correspondence between the abstract and the results.
Most of the concerns I raised previously have been addressed in this version. Before publication in Biomolecules, however, I suggest a few minor adjustments to further refine the manuscript:
1- move the data on pATM from supplementary to the main figures
2-line 238: remove the underlining of pATM
Author Response
Reviewer 1 :
We thank the reviewer for his/her comments
The suppl data have been included in the main text
The typo of line 238 has been corrected
Reviewer 2 Report
Comments and Suggestions for Authors
Accept in present form the author have significantly improved the manuscript
Author Response
Reviewer 2 :
We thank the reviewer for his/her comments.